# Analysis of Kerf Quality Characteristics of Kevlar Fiber-Reinforced Polymers Cut by Abrasive Water Jet

**DOI:** 10.3390/ma16062182

**Published:** 2023-03-08

**Authors:** Dinu-Valentin Gubencu, Carmen Opriș, Adelina-Alina Han

**Affiliations:** Mechanical Engineering Faculty, Politehnica University of Timisoara, 300222 Timisoara, Romania; dinu.gubencu@upt.ro

**Keywords:** Kevlar fiber-reinforced polymers, abrasive water jet cutting, full factorial experiment, surface roughness, kerf taper angle

## Abstract

Abrasive water jet machining has become an indispensable process for cutting Kevlar fiber-reinforced polymers used in applications such as ballistics protection, race cars, and protective gloves. The complex and diffuse action of a large number of input parameters leads to the need to evaluate the quality characteristics of the technological transformation as a result of the deployment of experimental studies adapted to the specific processing conditions. Thus, the paper focuses on identifying the influence of different factors and modeling their action on the characteristics that define the quality of the cut parts, such as the kerf taper angle and the *Ra* roughness parameter, by applying statistical methods of design and analysis of experiments.

## 1. Introduction

Kevlar^®^ is an organic fiber belonging to the aromatic polyamides (aramids) family. It was developed in 1965 by the scientists from DuPont. Due to their exceptional characteristics, such as high specific strength, excellent fatigue resistance, good chemical behavior with respect to fuels, vibration damping, and thermal stability, these fibers became an adequate option as composite reinforcements for a large number of high-demanding industrial applications [1,2].

Currently, there are many variants of fibers in the Kevlar^®^ family with different combinations of properties, which makes them suitable to meet various requirements. For instance, Kevlar^®^ 29 (K-29) has a high toughness and low stiffness, and is mainly used in industrial applications such as cables, brake linings, boats hull reinforcement, asbestos replacement, or protective armors [2,3]. Regarding antiballistic applications, K-29 is used for both civil and military purposes and to manufacture panels for lightweight military vehicles, pilot seats, or body vests. These applications require the absorption of a bullet or projectile kinetic energy, avoiding penetration and back face deformation of the armor and, thus, any major injury to the person [4].

Some of the important limitations of Kevlar^®^ fibers are their weakness in compression, poor resistance to UV rays, and hygroscopicity [2,5]. Among the disadvantages of Kevlar fiber-reinforced polymers (KFRP) are their low adhesion with impregnating resins and difficult machining by means of classical processes using hand-held tools. These processes have low reproducibility and accuracy and involve important tool wear. As a consequence, new technologies, such as abrasive water jet machining (AWJM), are widely used for cutting composite materials, including KFRP. However, compared with traditional cutting processes, the material treatment by AWJM is much more elaborate [6]. This cost-effective and flexible technique can achieve cutting of complicated shape parts with high precision and low residual stresses in the material structure. Although during the process no dust is generated in the ambient air, there are some environmental concerns related to loud background noise, unclean working areas, and high abrasive wear [7].

Basically, abrasive water- et cutting (AWJC) consists of using thin water jets under high pressure which pass through a convergent shape nozzle (Figure 1), determining the conversion of pressure energy into the kinetic energy of water [8]. Then, a high-speed water jet passes through a mixing chamber which is directly connected to the water nozzle. A stream of abrasive particles is added into the mixing chamber where high-energy water transfers onto the abrasive particles [9], accelerating them by the momentum exchange [10].

Following this, a mixture of water and abrasive particles produces a high coherent jet that passes through a focusing tube nozzle, which is directed towards the working area to cut the target material by means of erosion. When the induced stress surpasses the ultimate shear stress of the material, small chips of the material are removed [8]. The cutting head is relatively moved to the workpiece depending on the process type operated in order to expose new material to be machined.

The quality characteristics of the kerf after AWJC of composite materials can be estimated by analyzing several parameters, such as surface roughness, surface waviness (Figure 2a), kerf top width, *k_t_*, kerf bottom width, *k_b_*, and kerf taper angle, *k_a_* (Figure 2b).

Several studies have highlighted that under an initial damage zone caused by jet expansion before impact and the radial variation of jet energy, the cut surface can be separated into two regions: the smooth (cutting wear) zone or top zone and the rough (deformation wear) zone or bottom zone (Figure 2a) [11,12,13].

The cutting process occurs at the top zone of the machined surface under conditions in which most of the particles have a higher kinetic energy than the required destruction energy of processed material, resulting a smooth surface [8]. The higher roughness of the bottom zone is explained by the loss of jet penetration energy, which provokes its deflection and, as a result, profile waviness [11].

The kerf taper angle is influenced by the cutting capacity of the jet, so a higher kinetic energy increased by water jet pressure and a small increase in the abrasive mass flow rate improve this output parameter [14].

As the first step of the conducted research, a fishbone diagram was constructed, as shown in Figure 3, in order to organize the different categories of causes that can influence the quality characteristics of kerf obtained by AWJC. Some of the factors listed in the cutting method category and in subcategories, such as hydraulic system, abrasive system, and mixing system, can be easily controlled.

Therefore, numerous studies have focused on analyzing and quantifying the influence of these input parameters on surface roughness and taper angle of the kerf achieved by AWJC.

Siddiqui and Shukla carried out experimental research on AWJC of Kevlar epoxy composites, applying the Taguchi method and response surface methodology. They agreed, for the selected experimental range of each input parameter, that high water jet pressure, low abrasive flow rate, and low traverse speed ensure an optimum surface finish [11].

Sambruno et al. reported after performing an ANOVA analysis that when processing thermoplastic CFRP the slot walls become more vertical at a high water pressure and a low traverse rate [14].

Studying AWJM of carbon epoxy composites, Dhanawade and Kumar concluded that a medium value of the traverse speed, namely *v* = 100 mm/min in the experiment, ensures an acceptable surface roughness and high productivity [15]. They also observed the slight influence of the abrasive flow rate on the surface roughness and recommended a decrease in this parameter in order to reduce abrasive consumption.

Doreswamy et al. investigated composites with different reinforcements and established that kerf width is larger with the increase in water pressure and the stand-off distance and with the decrease in traverse speed. Abrasive flow rate had a marginal effect on kerf width [16]. Abidi et al. reported after analyzing AWJM of woven fabric CFRP that minimum surface roughness (*Ra* = 3.21 μm) was achieved for medium traverse speed (*v* = 2000 mm/min) and abrasive flow rate (*q* = 300 g/mm), associated with a minimum stand-off distance (*h* = 2 mm). On the other hand, kerf taper increased with the growth of both traverse speed and abrasive flow rate [17].

Azmir et al. [18] explored AWJC of Kevlar–phenolic composites using Taguchi’s experimental design and reached the conclusion that both investigated output parameters, surface roughness *Ra* and kerf taper ratio, decreased when reducing the traverse speed and stand-off distance. They also determined that increasing the hydraulic pressure resulted in lower values of those parameters, while abrasive flow rate showed no effect.

After carrying out an experimental program with five input parameters by means of Taguchi’s orthogonal array, Vikas and Srinivas [19] found that when processing glass epoxy composites a lower kerf taper angle can be achieved by increasing water jet pressure and decreasing grain size.

Shanmugam and Masood [20] conducted experimental research on AWJC of graphite epoxy and glass epoxy composites and decided that a combination of high water pressure, low traverse speed, and low standoff distance ensures the minimization of kerf taper angle.

Sathishkumar et al. [21] carried out an experimental study on AWJC of basalt–Kevlar–glass fiber-reinforced epoxy laminates, aiming to optimize its output parameters. The results showed that input parameters, such as traverse speed, abrasive mass flow rate, and standoff distance, generated an opposite effect on kerf quality characteristics. Thus, surface roughness decreased, and kerf taper angle increased when either the abrasive flow rate increased, or the traverse speed or standoff distance decreased.

Investigating AWJC of other hybrid composite laminates, consisting of basalt fiber and different loadings of fly ash particles as reinforcements for a vinyl ester polymer matrix, Ramraji et al. [22] concluded that high water jet pressure, low traverse speed, and standoff distance is recommended to reduce both kerf taper and surface roughness.

Many studies have been dedicated to the study of factors effects on the kerf surface finish when AWJC is conducted on other materials with high mechanical properties. Thus, in cutting Hardox 500 steel, Perec found an experimental model for the *Sq* roughness parameter [23], considering as input parameters traverse speed, abrasive flow rate, and water pressure, by means of a response surface methodology. While the pressure variation between 350 and 400 MPa had a statistically insignificant effect on *Sq*, the speed increase between 100 and 300 mm/min exerted the greatest influence on the roughness of the cut surface, causing its growth. Conversely, changing the flow rate to between 250 and 450 g/min resulted in an improvement in the *Sq* parameter.

The same three input parameters set at different levels were selected by Gupta et al. for analyzing kerf top width and kerf taper angle in the case of AWJC of marble [24]. Traverse speed exerted the most significant effect on top kerf width, followed by water pressure. For achieving minimum top kerf width, the settings of these two process parameters must be at the highest levels of 100 mm/min and 340 MPa, respectively. The kerf taper angle was significantly affected only by transverse speed and its minimum value was obtained at the lowest level of travel speed adopted in the experiment, i.e., 50 mm/min.

Major challenges for AWJC processes are to minimize both surface finish and kerf taper. The influence of factors and the results obtained in all previously presented studies are valid only under the particular conditions of the experiments performed, depending on the type and thickness of the material investigated or the values assigned to input variables. Thus, in order to choose appropriate combinations for the input parameters that are useful for achieving effective processes, a customized study on AWJC of KFRP composites must be conducted. Therefore, the objectives pursued in this work were to experimentally determine the mathematical link between the quality characteristics of AWJC of KFRP and the process parameters, adjusted between limits of technological interest for industrial applications.

## 2. Materials and Methods

The experimental approach focuses on modeling the process based on measured data, starting from an input–output model similar to a black box. That is, the in-depth analysis of the physical phenomena occurring in the workspace is deliberately neglected, aiming only to establish the link between the investigated objective functions and the influence factors using mathematical statistics.

The experimental program was designed using a full factorial experiment applied using 4 relevant influence factors with the purpose of finding the regression models.

### 2.1. Material

The specimens to be analyzed in this work (Figure 4) were AWJ cut into a rectangular shape of 60 mm × 30 mm, from a vacuum-bagged and autoclave-cured Kevlar epoxy laminate. This material was processed by Duqueine Composites for ballistic applications, resulting in a 9.5 mm thickness laminate formed by 18 layers of prepregs with 40% resin content. For prepreg reinforcement, woven fabric was used with the specifications presented in Table 1 [25]. Fabrics, which comprise of at least 2 woven-together threads (the warp and the weft) are available in several styles, selected according to necessary crimping and drapeability [26]. The tested material used a plain weave style, with each warp strand floating over and then under one fill strand, suitable for flat surface applications because the lay-up process over complex shaped molds is quite difficult [27].

### 2.2. Machining Equipment

The process was conducted on a JEDO CNC water jet abrasive machine (Figure 5). The equipment includes the following component units [8,9,10,11,12]:Water supply systemHigh-pressure water generatorAbrasive delivery systemCutting headCNC routerCatcher and drain system.

**Figure 5 materials-16-02182-f005:**
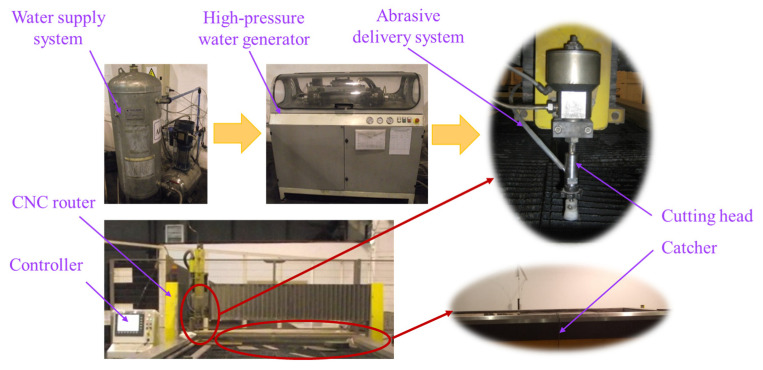
JEDO AWJC equipment setup.

The water supply system is basically a high-pressure water pump system including 2 different storage tanks: the cutting tank and the cooling water tank.

The high-pressure generating system includes 2 main units, the intensifier and the accumulator. The intensifier, acting as an amplifier, provides ultra-high pressurized water at approximately 200–400 MPa. The accumulator temporarily stores the energy of high-pressure water until is required. In order to prevent the fragmentation of the water jet, long-chain polymers are added to the water as stabilizers.

The abrasive delivery system consists of an abrasive hopper and a pneumatically operated valve to control the abrasive mass flow rate. The abrasive material typically used in most applications is garnet, which has a hardness of 8 on the Mohs scale [8]. The GMA garnet originated from Australia and was used for the investigations. This product presents some advantages over other abrasives, as it is totally natural, chemically inert, and free of any toxic metals or crystalline silica. The GMA garnet characteristics and composition [28] are presented in Table 2 and Table 3, respectively.

The cutting head, with the structure shown in Figure 1, is the equipment component where the mixing of high-pressure water and abrasive material occurs, thus achieving material removal as explained previously. The motion of the cutting head is controlled by a CNC router to desired coordinates of the working area, ensuring flexibility, productivity, and high dimensional precision, as well as accuracy of the machining process.

A catcher tank supports the rigid frame worktable and collects the pressurized water after cutting, thus dissipating energy and reducing noise.

### 2.3. Objective Functions and Influence Factors

The characterizations of the processed parts were assessed by determining 3 objective functions:arithmetic average roughness, *Ra_t_*, at the top zonearithmetic average roughness, *Ra_b_*, at the bottom zonekerf taper angle, *k_a_*, measured for the longer sides of the specimens.

The *Ra* roughness parameter was chosen because it is the most widely used as a global evaluation of the roughness amplitude on a profile and, also, is defined similarly in all standards, as the arithmetic mean of the absolute ordinate values Z(*x*), within the sampling length, *l_r_*, numerically equal to the characteristic wavelength of the profile filter *λ_c_*, which suppress the longwave component [29]:(1)$$Ra=\frac{1}{l_{r}}\;\int_{0}^{l_{r}}\left|\;Z\left(x\right)\right|\;dx$$
where Z(*x*) represents the heights of the assessed profile at any position *x.*

The kerf taper angle, *k_a_*, was calculated based on the difference between the measured top and bottom sides of the processed specimens, *b* and *B*, respectively, and the measured thickness of the specimen, *h* (Figure 3), using the standard formula:(2)$$k_{a}{=\tan}^{-1}\frac{B\;-\;b}{2\;h}\;\times \;\frac{180{}^{\circ}}{\pi }$$

Input parameters were selected to test their influence not only on the quality characteristics of the cut, but also on the efficiency of KFRP processing process, the analysis of which was the objective of a separate study.

Thus, 4 influence parameters were chosen for carrying out the experimental programme:traverse speed, *v* (mm/min)focusing tube diameter, *D* (mm)abrasive flow rate, *q* (g/min)abrasive grain size, *g* (mesh #)

The experimental range for each factor (Table 4) was selected taking into account not only previous conducted studies by several researchers and industrial expertise, but our own preliminary experiments. At the same time, some limitations regarding the equipment, which is supplied with 2 focusing tubes each with a focusing tube diameter of 0.76 mm or 1.00 mm, determined the levels for this input parameter. The abrasive flow rate levels were selected in correlation with the other process parameters to avoid saturation problems and difficulty in the transfer of momentum to the abrasive particles.

The initial parameter settings, which were kept unchanged during all experimental trials, are presented in Table 5. Their values were selected considering the presumed positive effect on the objective functions.

## 3. Results and Discussions

For modeling the action of a number of *p* influence factors selected, *x_j_*, on a performance characteristic, *y*, by means of a 2*^p^* full factorial experiment, the experimental results are used to find the constants, *b*_0_, *b_j_*, *b_jk_*, of the polynomial [30]:(3)$${y=b}_{0}+\overset{\;p}{\underset{\;j=1}{\sum }}b_{j}{\;x}_{j}+\overset{\;p}{\underset{\;j,k=1,\;j\neq k}{\sum }}b_{jk}{\;x}_{j}{\;x}_{k}$$

This experimental design ensures that a high precision model is obtained with a minimum number of trials, thus reducing the investigation costs.

### 3.1. Experimental Design and Measured Results

The experimental matrix (Table 6) was constructed according to the principles of designing full factorial experiments, containing all the possible combinations of the factors levels [30]. The method selected for estimating the experimental error was to replicate each run, *i*, of the experiment. Table 6 presents results for all three objective functions investigated.

For processing, analyzing, and plotting the experimental data, the statistical dedicated software Minitab^®^ 17 was utilized.

### 3.2. Analysis of the Surface Roughness

Surface roughness, namely the *Ra* parameter, was measured using a handheld Taylor Hobson Surtronic 25 stylus tester (Figure 6). This parameter was established separately for the smooth and the rough zone of the kerf, *Ra_t_* and *Ra_b_*, respectively (Figure 2a), adopting a unitary procedure for both investigated areas by keeping the most unfavorable value among those determined after performing two measurements on each longer side cut surface of the samples. For all experimental trials, stylus traverse movements perpendicular to the lay direction were measured and the Gaussian filter was applied.

The cutoff length values were selected according to standard recommendations [31] as follows: *λ_c_* = 2.5 mm, if *Ra* < 10 μm or *λ_c_* = 8 mm, if *Ra* > 10 μm. On the other hand, there were differences between the initial settings for the two kerf zones regarding the number of sampling lengths considered to establish the evaluation lengths. Thus, this number for the kerf top zone was *N_top_* = 5, but for the bottom zone, characterized by insufficient space on the sample surface, the agreed value was *N_bottom_* = 2.

The effects of selected input parameters on the top zone roughness, *Ra_t_*, are shown in Figure 7a and on the bottom zone roughness, *Ra_b_*, in Figure 7b. From both figures, it can be observed that increasing the abrasive flow rate, *q*, and mesh #, *g*, ensures a lower surface roughness; meanwhile, increasing the other two factors, traverse speed, *v*, and focusing tube diameter, *D*, have an opposite effect.

The positive effect of *q* growth can be clarified by the proximity of the trajectories of the abrasive particles that exert the erosive effect on the surface of the cut. The higher the number of particles involved, the higher the number of initial geometrically superposed craters formed by the particles at the impacted surface. These craters will initiate closer micro-cracks resulting in a better surface finish. One of the micro-mechanisms involved in material separation, plastic deformation, results in the appearance of traces on the kerf surface and the sizes are dependent on the abrasive grain size, thus explaining the decrease in roughness with a higher mesh # of the sand.

On the other hand, a lower *D* value leads to a reduction in the impact surface of the jet on the workpiece and more close trajectories of the abrasive particles are generated, causing a decrease in roughness parameter *Ra* in both zones. A similar effect on particle paths is caused by the decreases of the traverse speed, *v*, combined with a smaller jet deflection, explaining the improvement of surface finish.

An analysis of variance (ANOVA) was carried out to distinguish the statistically significant effects on top surface roughness, *Ra_t_* (Table 7), and on bottom surface roughness, *Ra_b_* (Table 8).

Since the *p*-value is less than 0.05 for all selected influence factors, these have a statistically significant effect both on *Ra_t_* and *Ra_b_* at the 95% confidence level. Contrarily, since the *p*-value is higher than 0.05 (with one exception) the interaction effects are insignificant; as a result, they can be eliminated from the future model.

Consequently, the regression equations of the fitted models developed for coded values of input variables are:*Ra_t_* = 10.031 + 1.450 *v* + 1.056 *D* − 0.944 *q* − 0.606 *g*
(4)
*Ra_b_* = 15.025 + 2.013 *v* + 1.600 *D* − 1.306 *q* − 1.069 *g* + 0.612 *v D*(5)

By examining the regression coefficients values it can be seen that all the input parameters have a higher effect on *Ra_b_* than on *Ra_t_.* Furthermore, the hierarchy of factors, in order of their influence on surface roughness, is the same for the top and bottom zones (Figure 8).

The normal probability plot (Figure 9) is a scatter plot with the theoretical percentiles of the normal distribution on the *x*-axis and the sample percentiles of the residuals on the *y*-axis, used to test the normal distribution of residuals. The residuals, represented on the plot with blue dots, are placed on a straight line for both response functions, showing that the relationships between the theoretical percentiles and the sample percentiles are approximately linear and proving that errors are indeed normally distributed.

Figure 10 illustrates the agreement between the experimental data (represented in the graph with red squares) and the predicted values, confirming the consistency of data for both investigated roughness zones.

The empirical models can be graphically represented considering various combinations of two factors as independent variables and setting the other factors at different constant values. As an example, three-dimensional graphs of both fitted models with natural values of the factors are presented in Figure 11. Figure 12 shows the contour lines of constant responses *Ra_t_* and *Ra_b_* in the selected two factors plane.

### 3.3. Analysis of the Kerf Taper Angle

The kerf taper angle, *k_a_*, was evaluated by carrying out two measurements for each specimen with a FARO-Edge arm and retaining for the analysis the higher value (Table 6). The main effects of the influence factors were plotted in Figure 13a, showing that the increase in traverse speed, *v*, and the focusing tube diameter, *D*, leads to a higher kerf taper angle, *k_a_*. Contrarily, the rise in abrasive flow rate, *q*, and in mesh number, *g*, generates the decrease in *k_a_*. The magnitude of factor influence allowed their ranking in Figure 13b.

The unfavorable effect on *k_a_* of traverse speed increasing can be explained by the reduction in the overlap of abrasive grain impacts and, as a consequence, of its cutting capacity. On the other hand, increasing *q* in the selected experimental range avoids collisions between abrasive particles and its edge rounding and determines an improvement of *k_a_* due to the greater cutting capacity of the jet.

A common, simplified interpretation of the effects of the input parameters on the investigated quality characteristics can be formulated after the analysis of the physical phenomena that occur in the working space. Material removal mechanisms involved in AWJC of fiber-reinforced polymers consist of both erosive processes of the matrix and shearing and pulling-out processes of the fibers [32]. Basically, these processes depend on the amount of kinetic energy in the solid particles at impingement, following the rule that an increase in the energy level leads to an improvement in the quality characteristics of the kerf, i.e., to lower roughness and taper angle.

Thus, an increase in the traverse speed reduces the kinetic energy per unit of time and a higher focusing tube diameter diminishes the kinetic energy density of the jet upon impact on the processed material. As a result, a higher surface roughness and kerf taper angle are obtained. Contrarily, when the abrasive flow rate and abrasive mesh # are increased, more particles strike the target area. Therefore, the jet will gain a higher kinetic energy that enhances its penetration capacity of the workpiece. As a consequence, smoother cut surfaces will be obtained and the top and bottom widths of the kerf will have close values, determining a smaller taper angle.

An ANOVA was performed to analyze the statistical significance of input parameter effects and of their interactions with the kerf taper angle (Table 9).

Analyzing the *p*-value column proves that all factor effects are significant at the 95% confidence level, but the interaction effects are not, so they will be neglected in the regression model:*k_a_* = 1.2097 + 0.1866 *v* + 0.1322 *D* − 0.1384 *q* − 0.0709 *g*(6)

Figure 14a reveals that the residuals, represented in the graph by blue dots, are positioned quite close to a straight line, attesting the normal distribution of errors. Figure 14b demonstrates good concordance between measured kerf taper angles, represented in the graph by red squares, and predicted values by the fitted model.

The empirical models can be graphically represented considering various combination of two factors as independent variables and setting the other factors at different constant values. An example of a response surface and contour plot for kerf taper angle, *k_a_*, is presented in Figure 15. Contour plots in particular are of practical importance, allowing the selection of appropriate combinations of factor levels that ensure the fulfillment of specifications related to the objective function, in this case, kerf taper angle.

### 3.4. Investigations on Delamination

Visual and macroscopic examinations of the samples were conducted in order to emphasize the kerf aspect. An Olympus stereo microscope type SZX7 with Analysis 5.0 software was used for sample imaging.

Almost all samples showed a cut surface without delamination damage, unlike the following cases, where some problems occurred. For instance, sample 11 presented delamination and broad and accentuated kerf (Figure 16).

The analysis revealed for sample 18 a discreet kerf with a minor defect at the start point (see detail in Figure 17a).

On the other hand, many samples, as seen in Figure 17b for sample 13, exhibited minor fiber pullouts at the exit of the abrasive water jet.

As a result of this analysis, a causal relationship could not be identified between the levels assigned to the factors in the experiment and the occurrence of these isolated cases of sample delamination.

## 4. Conclusions

Following the experiment carried out, the most important conclusions regarding AWJC of KFRP that can be drawn are:All the input parameters adjusted in the experiment proved to have a significant influence on the three analyzed quality characteristics.The study revealed that traverse speed had the greatest influence on surface roughness, both on the top cutting zone, *Ra_t_,* and on the bottom cutting zone, *Ra_b_*, followed by focusing tube diameter, abrasive flow rate, and abrasive grain size, but the amplitude of the effects produced by these factors is magnified at the bottom zone.The traverse speed had the most important influence on the kerf taper angle, and the hierarchy was continued by abrasive flow rate, focusing tube diameter, and abrasive grain size.Experimental models with coefficients of determination at approximately 90% were found to describe the action of selected factors on surface roughness *Ra_t_* (R-sq = 90.60%), *Ra_b_* (R-sq = 91.03%), and kerf taper angle *k_a_* (R-sq = 89.18%).The increase in abrasive flow rate and in mesh number had a positive effect, determining the decrease in all three investigated quality characteristics, while the growth in traverse speed and focusing tube diameter generated a negative effect.Therefore, the optimum combination of factor levels that assures minimum surface roughness and kerf taper angle under the conditions of the experiment performed is: *v* = 100 mm/min, *D* = 0.76 mm, *q* = 200 g/min, *g* = 200 mesh #.

## Figures and Tables

**Figure 1 materials-16-02182-f001:**
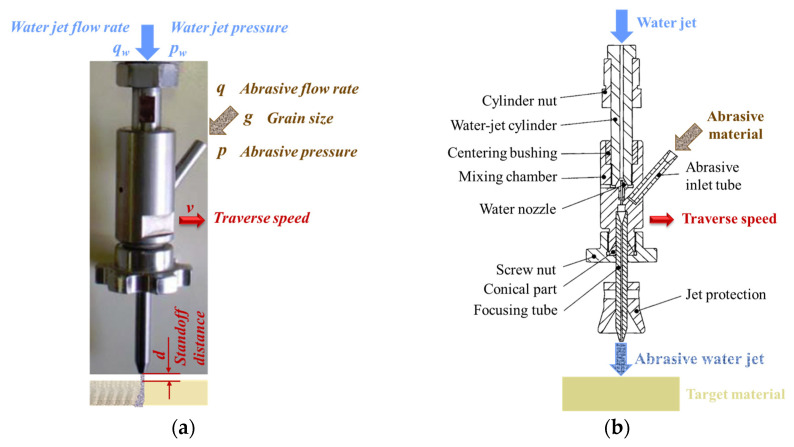
Principle of AWJC: (**a**) process parameters (**b**) components of cutting head.

**Figure 2 materials-16-02182-f002:**
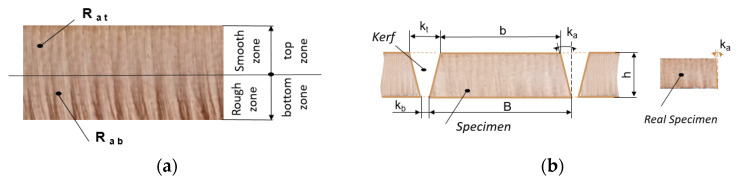
Quality characteristics of cut surfaces: (**a**) roughness zones on KFRP (**b**) dimensional parameters of the kerf. (b represents the specimen top width and B represents the specimen bottom width).

**Figure 3 materials-16-02182-f003:**
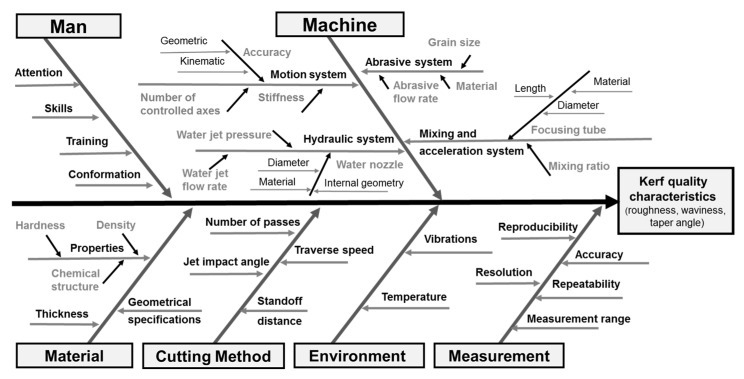
Cause–effect diagram for AWJC.

**Figure 4 materials-16-02182-f004:**
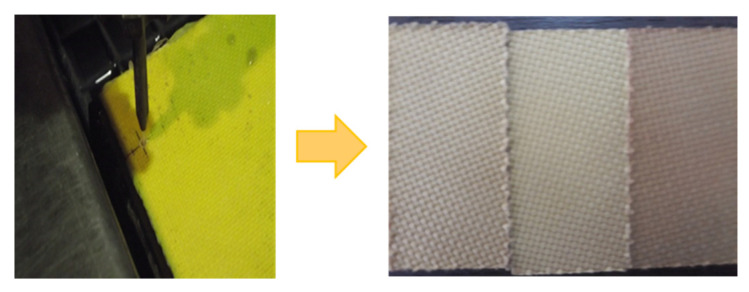
KFRP specimens cut by AWJC.

**Figure 6 materials-16-02182-f006:**
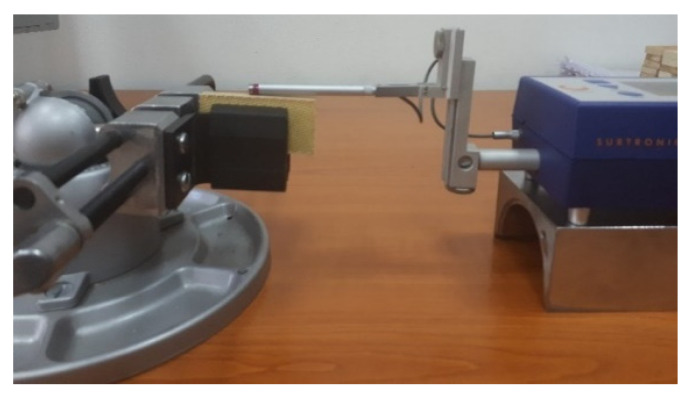
Experimental measuring of the surface roughness.

**Figure 7 materials-16-02182-f007:**
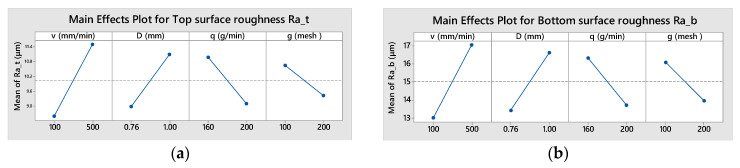
Main effects plot: (**a**) for top surface roughness, *Ra_t_* (**b**) for bottom surface roughness, *Ra_b_*.

**Figure 8 materials-16-02182-f008:**
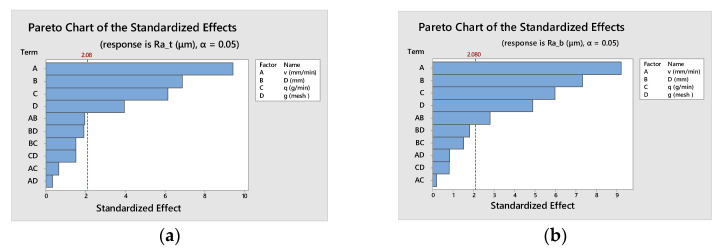
Hierarchy of factors effects: (**a**) for top surface roughness, *Ra_t_* (**b**) for bottom surface roughness, *Ra_b_*.

**Figure 9 materials-16-02182-f009:**
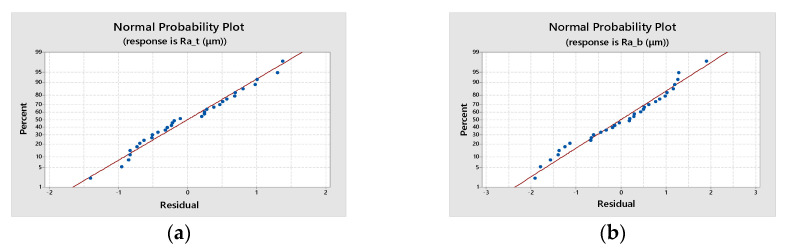
Normal probability plot: (**a**) for top surface roughness, *Ra_t_* (**b**) for bottom surface roughness, *Ra_b_*.

**Figure 10 materials-16-02182-f010:**
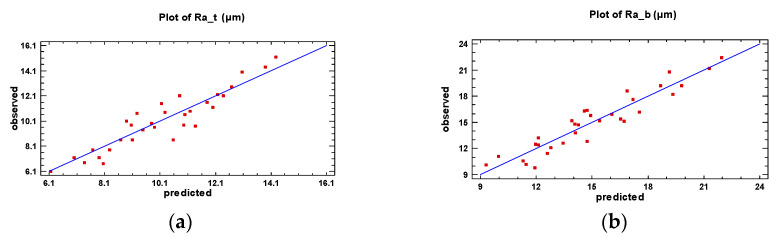
Comparison of experimental and model predicted surface roughness: (**a**) for top surface roughness, *Ra_t_* (**b**) for bottom surface roughness, *Ra_b_*.

**Figure 11 materials-16-02182-f011:**
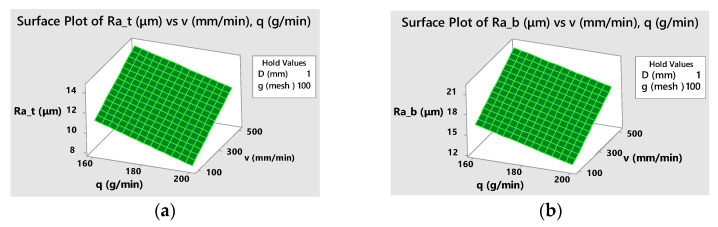
Response surfaces of roughness versus traverse speed and abrasive flow rate at *D* = 1 mm, *g* = 100: (**a**) for top surface roughness, *Ra_t_* (**b**) for bottom surface roughness, *Ra_b_*.

**Figure 12 materials-16-02182-f012:**
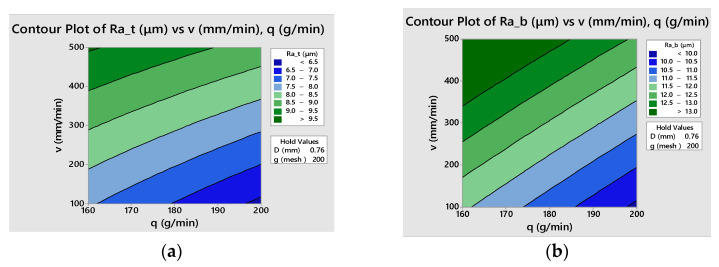
Contour plots of surface roughness versus traverse speed and abrasive flow rate at *D* = 1 mm, *g* = 100: (**a**) for top surface roughness, *Ra_t_* (**b**) for bottom surface roughness, *Ra_b_*.

**Figure 13 materials-16-02182-f013:**
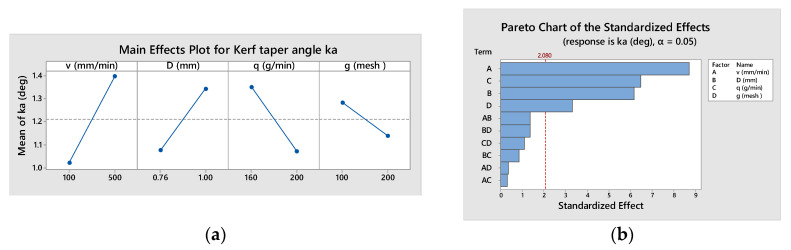
Effects on kerf taper angle, *k_a_*: (**a**) main effects plot (**b**) effects hierarchy.

**Figure 14 materials-16-02182-f014:**
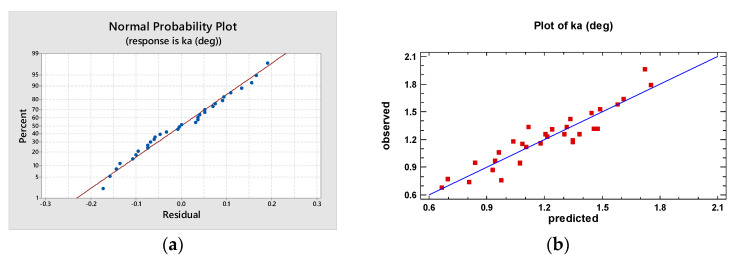
Diagnostic plots for kerf taper angle, *k_a_*: (**a**) normal probability plot (**b**) observed versus predicted values.

**Figure 15 materials-16-02182-f015:**
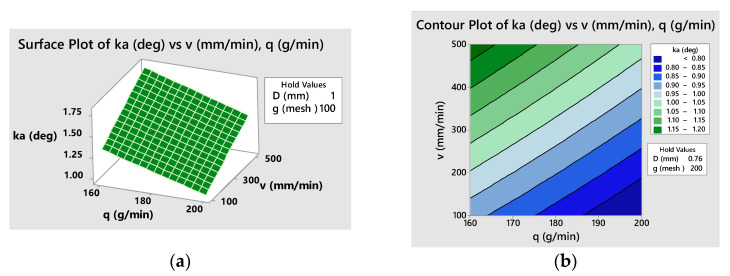
Response plots for kerf taper angle, *k_a_*, versus traverse speed and abrasive flow rate at *D* = 1 mm, *g* = 100: (**a**) 3D surface plot (**b**) contour plot.

**Figure 16 materials-16-02182-f016:**
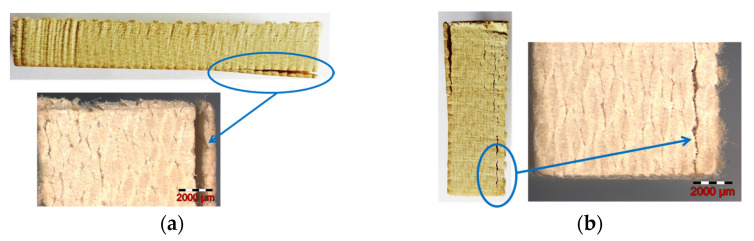
Sample 11: (**a**) length surface showing delamination and broad kerf (**b**) width surface showing delamination.

**Figure 17 materials-16-02182-f017:**
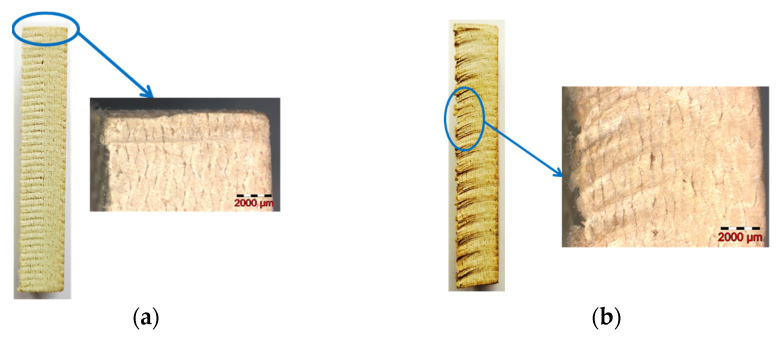
Kerf aspect: (**a**) sample 18 (**b**) sample 13.

**Table 1 materials-16-02182-t001:** Fabric specifications.

Characteristic	Specification
Composition	100% para-aramid Kevlar 29
Warp	3300 dtex
Weft	3300 dtex
Thread count	6.65 × 6.65 ± 0.20 ends/cm
Density	440 ± 10 g/m^2^
Tensile strength	2863 MPa
Tensile modulus	67 GPa
Elongation at break	3.7%
Weave	Plain
Treatment	Washed

**Table 2 materials-16-02182-t002:** Physical characteristics of GMA garnet (typical).

Property	Value
Bulk Density	2.3 T/m^3^
Specific Gravity	4.1
Hardness (Mohs)	7.5–8.0
Melting Point	1250 °C
Shape of natural grains	Sub-angular

**Table 3 materials-16-02182-t003:** Composition of GMA garnet (typical).

Mineral and Metal Composition		Average Chemical Composition	
Garnet (Almandite)	97–98%	SiO_2_	36%
Ilmenite	1–2%	Al_2_O_3_	20%
Zircon	0.2%	FeO	30%
Quartz (free silica)	<0.5%	Fe_2_O_3_	2%
Others	0.25%	TiO_2_	1%
Ferrite (free iron)	<0.01%	MnO	1%
Lead	<0.002%	CaO	2%
Copper	<0.005%	MnO	6%
Other Heavy Metals	<0.01%		

**Table 4 materials-16-02182-t004:** Experimental levels and ranges of variation for the influence factors.

Factors	Symbol	Central Point 0	Range of Variation Δ _j_	Lower Level −1	Higher Level +1
Traverse speed, *v* (mm/min)	*x_1_*	300	200	100	500
Focusing tube diameter, *D* (mm)	*x_2_*	–	0.12	0.76	1.00
Abrasive flow rate, *q* (g/min)	*x_3_*	180	20	160	200
Abrasive grain size, *g* (mesh #)	*x_4_*	150	50	100	200

**Table 5 materials-16-02182-t005:** AWJC process experimental settings.

Parameter	Specification
Water pressure, pw (MPa)	300
Water nozzle diameter, d0 (mm)	0.33
Standoff distance, d (mm)	3
Jet angle of attack, γ (deg.)	90
No of passes (–)	1
Abrasive material	GMA garnet

**Table 6 materials-16-02182-t006:** Matrix and results of the 2^4^ full factorial experiments.

Run No *i*	A: *v*		B: *D*		C: *q*		D: *g*		*Ra_t_* (μm)		*Ra_b_* (μm)		*k_a_* (deg)	
	Coded	(mm/min)	Coded	(mm)	Coded	(g/min)	Coded	(mesh #)	*Ra_t1_*	*Ra_t2_*	*Ra_b1_*	*Ra_b2_*	*k_a1_*	*k_a2_*
1	−1	100	−1	0.76	−1	160	−1	100	10.7	10.1	14.8	16.4	1.15	1.34
2	+1	500	−1	0.76	−1	160	−1	100	12.2	11.6	18.6	16.2	1.32	1.53
3	−1	100	+1	1.00	−1	160	−1	100	9.7	10.6	15.9	15.1	1.19	1.26
4	+1	500	+1	1.00	−1	160	−1	100	15.2	14.4	21.2	22.4	1.96	1.79
5	−1	100	−1	0.76	+1	200	−1	100	6.8	7.2	10.2	12.4	0.74	0.95
6	+1	500	−1	0.76	+1	200	−1	100	10.8	9.6	14.7	15.8	1.16	1.23
7	−1	100	+1	1.00	+1	200	−1	100	9.4	8.6	12.6	13.8	0.94	1.12
8	+1	500	+1	1.00	+1	200	−1	100	12.1	11.2	19.2	18.2	1.49	1.32
9	−1	100	−1	0.76	−1	160	+1	200	6.7	7.8	9.8	11.4	0.97	0.76
10	+1	500	−1	0.76	−1	160	+1	200	9.8	8.6	12.8	15.2	1.34	1.17
11	−1	100	+1	1.00	−1	160	+1	200	11.5	9.9	15.2	16.3	1.26	1.31
12	+1	500	+1	1.00	−1	160	+1	200	14.0	12.8	20.8	19.2	1.58	1.64
13	−1	100	−1	0.76	+1	200	+1	200	6.1	7.2	10.1	11.1	0.68	0.77
14	+1	500	−1	0.76	+1	200	+1	200	9.8	8.6	13.2	12.1	1.18	0.95
15	−1	100	+1	1.00	+1	200	+1	200	7.8	7.2	10.6	12.5	0.87	1.06
16	+1	500	+1	1.00	+1	200	+1	200	10.9	12.1	15.4	17.6	1.26	1.42

**Table 7 materials-16-02182-t007:** ANOVA for top surface roughness, *Ra_t_*.

Source	DF	Adj SS	Adj MS	F-Value	*p*-Value
Model	10	152.707	15.2707	20.20	0.000
Linear	4	143.244	35.8109	47.47	0.000
V	1	67.280	67.2800	89.19	0.000
D	1	35.701	35.7012	47.33	0.000
q	1	28.501	28.5013	37.78	0.000
g	1	11.761	11.7613	15.59	0.001
Two-way interaction	6	9.464	1.5773	2.09	0.098
v × D	1	2.880	2.8800	3.82	0.064
v × q	1	0.320	0.3200	0.42	0.522
v × g	1	0.080	0.0800	0.11	0.748
D × q	1	1.711	1.7113	2.27	0.147
D × g	1	2.761	2.7613	3.66	0.069
q × g	1	1.711	1.7112	2.27	0.147
Error	21	15.841	0.7543		
Lack-of-Fit	5	7.681	1.5363	3.01	0.042
Pure error	16	8.160	0.5100		
Total	31	168.549			

**Table 8 materials-16-02182-t008:** ANOVA for bottom surface roughness, *Ra_b_*.

Source	DF	Adj SS	Adj MS	F-Value	*p*-Value
Model	10	325.248	32.525	21.31	0.000
Linear	4	302.678	75.669	49.58	0.000
V	1	129.605	129.605	84.91	0.000
D	1	81.920	81.920	53.67	0.000
q	1	54.601	54.601	35.77	0.000
g	1	36.551	36.551	23.95	0.000
Two-way interaction	6	22.570	3.762	2.46	0.058
v × D	1	12.005	12.005	7.87	0.011
v × q	1	0.061	0.061	0.04	0.843
v × g	1	1.051	1.051	0.69	0.416
D × q	1	3.511	3.511	2.30	0.144
D × g	1	4.961	4.961	3.25	0.086
q × g	1	0.980	0.980	0.64	0.432
Error	21	32.052	1.526		
Lack-of-Fit	5	11.232	2.246	1.73	0.186
Pure error	16	20.820	1.301		
Total	31	357.300			

**Table 9 materials-16-02182-t009:** ANOVA for kerf taper angle, *k_a_*.

Source	DF	Adj SS	Adj MS	F-Value	*p*-Value
Model	10	2.533	0.25331	17.32	0.000
Linear	4	2.447	0.61181	41.82	0.000
V	1	1.113	1.1137	76.13	0.000
D	1	0.559	0.5591	38.22	0.000
q	1	0.613	0.6132	41.92	0.000
g	1	0.161	0.1612	11.01	0.003
Two-way interaction	10	2.533	0.25331	17.32	0.464
v × D	6	0.085	0.0114	0.98	0.188
v × q	1	0.027	0.0270	1.85	0.762
v × g	1	0.001	0.0013	0.09	0.718
D × q	1	0.002	0.0019	0.13	0.398
D × g	1	0.011	0.0108	0.74	0.188
q × g	1	0.027	0.0270	1.85	0.285
Error	21	0.307	0.0146		
Lack-of-Fit	5	0.094	0.0188	1.41	0.272
Pure error	16	0.213	0.0133		
Total	31	2.8403			

## Data Availability

Not applicable.

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
