# Peer review of "Analysis of Kerf Quality Characteristics of Kevlar Fiber-Reinforced Polymers Cut by Abrasive Water Jet"

_materials, 2023, doi:10.3390/ma16062182_

Round 1

Reviewer 1 Report

The paper focuses on identifying the influence of different factors and modeling their action on the characteristics that define the quality of the cut parts, such as kerf taper angle and Ra roughness parameter, by applying statistical methods of design and analysis of experiments. The paper needs some improvement before acceptance for publication. My detailed comments are as follows.

1) The effects of the input parameters on the roughness should be discussed more deeply. And the mechanism of the effects of the input parameters on the roughness should be analyzed more deeply.

2) The mechanical model of the cutting process should be established, which is the foundation to analyze the effects of the input parameters.

3) Please revise and polish introduction and sentences of this paper.

Author Response

1) The effects of the input parameters on the roughness should be discussed more deeply. And the mechanism of the effects of the input parameters on the roughness should be analyzed more deeply.

This paper constitutes only a preamble regarding this type of experiment, aiming to present the problem and first trials. The next paper will focus on a deeper analysis on the effect of the input parameters on the roughness. However, we provided additional interpretation about the input parameters effects.

2) The mechanical model of the cutting process should be established, which is the foundation to analyze the effects of the input parameters.

This paper was focused on identifying the influence of different factors and modeling their action on the characteristics that define the quality of the cut parts establishing an empiric model; the future research will be focused on establishing a mechanical model of the cutting process.

3) Please revise and polish introduction and sentences of this paper.

Thank you kindly for your observation, we have revised the introduction and uploaded a revised version of the paper.

Reviewer 2 Report

The authors presented abrasive water jet cutting of Kevlar fiber-reinforced polymers problems with identifying the influence of distinct factors and modeling their action on the characteristics that define the quality of the cut parts, such as cut kerf angle and Ra roughness parameter, by applying statistical methods of design and analysis of experiments.

The paper is remarkably interesting, and I have no fundamental doubt in the presented research.

I found some errors in this manuscript, and it must be improved.

Weak

The fundamental weakness is a mixture of US English and UK English. Authors should choose one of them (anyone) and stick to it. A mix of these languages is considered an error.

Noticed errors

1.       The analysis of the state of the issue is too narrow and is limited only to the AWJ cutting area of Kevlar composites. For a conference paper this might be enough, but for a research paper in a reputable journal it is far too little. In my opinion, this analysis should be supplemented with the scope of application of statistical methods in AWJ technology:

- Modeling of Abrasive Water Suspension Jet Cutting Process Using Response Surface Method

- Multiple Response Optimization of Abrasive Water Jet -Cutting Process using Response Surface Methodology (RSM)

- Experimental Research into Marble - Cutting by Abrasive Water Jet

- Efficiency of Tool Steel Cutting by Water Jet with Recycled Abrasive Materials

2.       In chapter 1. Introduction lacked a summary of the state of the issue, a clear presentation of the research gap and succinctly stated purpose of this paper.

3.       Fig.1 (and whole paper). In abrasive water-jet processing, it is better to adopt traverse speed in place of traverse rate to avoid confusion with abrasive (or water) flow rate.

4.       Fig. 2b. What do the b and B parameters mean, and for what purpose did the authors introduce them?

5.       In Chapter 2 is almost no information about the abrasive materials. Detailed characteristics of the abrasive material, its manufacturer, deposit location, type (alluvial or crushed rock), grain size distribution and physical and chemical properties should be given. A photograph of a typical single grain and multiple grains would also be helpful to compare grain shapes in the population.

6.       Among the information about the conditions for measuring roughness, the specification of the filters used is missing. Roughness measurement needs to add information on signal filtering and the filters used in the roughness measurements. I also did not find information on the length of the measurement section.

7.       Figs. 11, 12 and 15. Authors should decide on 1 type of chart. In my opinion, a 3D chart (surface chart) illustrates better than a 2D chart (contour chart) the variability of the effect depending on the control parameters, and I recommend such a change.

Small errors

These errors do not diminish the value of this interesting work, but need to be improved

1.       Pages 2/3, 6/7, 8/9. Bad pagination.

Author Response

Thank you kindly for your observation, we have revised the language and uploaded a revised version of the paper.

Noticed errors

  1. The analysis of the state of the issue is too narrow and is limited only to the AWJ cutting area of Kevlar composites. For a conference paper this might be enough, but for a research paper in a reputable journal it is far too little. In my opinion, this analysis should be supplemented with the scope of application of statistical methods in AWJ technology:

- Modeling of Abrasive Water Suspension Jet Cutting Process Using Response Surface Method

- Multiple Response Optimization of Abrasive Water Jet -Cutting Process using Response Surface Methodology (RSM)

- Experimental Research into Marble - Cutting by Abrasive Water Jet

- Efficiency of Tool Steel Cutting by Water Jet with Recycled Abrasive Materials

Thank you kindly for your observation, we extended the analysis of the state of the issue and included the specified topics.

  1. In chapter 1. Introduction lacked a summary of the state of the issue, a clear presentation of the research gap and succinctly stated purpose of this paper.

Thank you kindly for your observation, we have revised the introduction and uploaded a revised version of the paper.

  1. Fig.1 (and whole paper). In abrasive water-jet processing, it is better to adopt traverse speed in place of traverse rate to avoid confusion with abrasive (or water) flow rate.

We will modify figure 1 and adopt traverse speed in place of traverse rate for the whole paper.

  1. Fig. 2b. What do the b and B parameters mean, and for what purpose did the authors introduce them?

In figure 2b, b represents the width of the sample on the top surface, where the water jet penetrated the material. B represents the width of the sample on the opposite surface.

  1. In Chapter 2 is almost no information about the abrasive materials. Detailed characteristics of the abrasive material, its manufacturer, deposit location, type (alluvial or crushed rock), grain size distribution and physical and chemical properties should be given. A photograph of a typical single grain and multiple grains would also be helpful to compare grain shapes in the population.

Information regarding the abrasive material, disclosed by Duqueine Composites, are provided in the revised version of the paper.

  1. Among the information about the conditions for measuring roughness, the specification of the filters used is missing. Roughness measurement needs to add information on signal filtering and the filters used in the roughness measurements. I also did not find information on the length of the measurement section.

Measuring conditions were:

- Cutoff (sampling) lengths: λc = 2.5 mm, if Ra < 10 μm or λc = 8 mm, if Ra > 10 μm.;

- No. of sampling lengths: Ntop = 5 and Nbottom = 2 for kerf top zone and bottom zone, respectively;

- Gaussian filter for all measurements.

  1. Figs. 11, 12 and 15. Authors should decide on 1 type of chart. In my opinion, a 3D chart (surface chart) illustrates better than a 2D chart (contour chart) the variability of the effect depending on the control parameters, and I recommend such a change.

We agree that the 3D-chart illustrates better the variability of the effects, but we think that the 2D-chart has a higher practical importance, allowing an easy selection of processing parameters combinations that ensure the required quality specifications. So, we would like to keep both representation types.

Small errors

These errors do not diminish the value of this interesting work, but need to be improved

  1. Pages 2/3, 6/7, 8/9. Bad pagination.

Thank you kindly for your observation, we have revised the pagination and uploaded a revised version of the paper.

Reviewer 3 Report

The manuscript presents AWJM machining of Kevlar-epoxy 147 laminate, and the manuscript is generally well-structured. However, the following aspects need to be taken care of for improving the quality-

1. Did the authors use any additive stabilizers for the stability of the water jet? Was there any bubble-formation or cavitation noticeable in the water jet at different pressures?

2. Since the kerf surfaces had lay-marks and directional features, the authors should consider other roughness parameters also for the study.

3. How did the authors ensure that the 2D roughness measures were conducted in a particular orientation with respect to the the directional features?

4. How does the the orientation of the kevlar fibers affect the kerf properties? Did the authors perform all the experiments for a specific orientation of the kevlar fibers?

5. How reproducible were the defects described in section 3.4?

6. The authors may consider expanding on the physico-mechanical explanation of the experimental observations.

5. 

Author Response

The manuscript presents AWJM machining of Kevlar-epoxy 147 laminate, and the manuscript is generally well-structured. However, the following aspects need to be taken care of for improving the quality-

  1. Did the authors use any additive stabilizers for the stability of the water jet? Was there any bubble-formation or cavitation noticeable in the water jet at different pressures?

We used additive stabilizers to ensure the stability of the water. Thus there was not any bubble-formation or cavitation noticeable in the water jet.

  1. Since the kerf surfaces had lay-marks and directional features, the authors should consider other roughness parameters also for the study.

Thank you kindly for your observation, future scientific papers will consider other roughness parameters.

  1. How did the authors ensure that the 2D roughness measures were conducted in a particular orientation with respect to the the directional features?

All the roughness measurements were carried out ensuring that the stylus traverse movements are perpendicular to the lay direction. Thus, for the kerf top zone, stylus travel direction was aligned parallel with the jet impact surface, considered as a datum feature. For the bottom zone, the orientation of the pattern was established by measuring its angle with a toolmaker microscope and then the sample was positioned at this angle using a rotary table.

  1. How does the the orientation of the kevlar fibers affect the kerf properties? Did the authors perform all the experiments for a specific orientation of the kevlar fibers?

The experiments did not take into account a specific orientation of the kevlar fibers. We provided additional information about the fabric weave style used by Duqueine Composites for processing the laminate and uploaded a revised version of the paper.

  1. How reproducible were the defects described in section 3.4?

The kerf aspects in section 3.4 were shown in order to exemplify typical defects that may occur during the cutting process. The pictures presented show the kerf aspect when a certain parameters combination is used and it is also specified that this was an isolated case.

  1. The authors may consider expanding on the physico-mechanical explanation of the experimental observations.

Thank you kindly for your observation, future scientific papers will consider this suggestion. However, we expanded our explanations in this paper too and uploaded a revised version of the paper.

Round 2

Reviewer 1 Report

After reviewing the manuscript, I think the manuscript proper to be accepted.

Reviewer 2 Report

The authors have corrected all errors and thus the obstacles to publishing this paper have disappeared.